# Novel Oncogenic Value of C10orf90 in Colon Cancer Identified as a Clinical Diagnostic and Prognostic Marker

**DOI:** 10.3390/ijms251910496

**Published:** 2024-09-29

**Authors:** Chuangdong Ruan, Yuqin Zhang, Daoyang Chen, Mengyi Zhu, Penghui Yang, Rongxin Zhang, Yan Li

**Affiliations:** Guangdong Provincial Key Laboratory of Advanced Drug Delivery, Guangdong Provincial Engineering Center of Topical Precise Drug Delivery System, Department of Biotechnology, School of Life Sciences and Biopharmaceutics, Guangdong Pharmaceutical University, Guangzhou 510006, China; dong123123122023@163.com (C.R.); 18984776137@163.com (Y.Z.); staog5174@gmail.com (D.C.); zhumengyihi@163.com (M.Z.); 17531087793@163.com (P.Y.)

**Keywords:** *C10orf90*, pan-cancer, COAD, comprehensive analysis

## Abstract

C10orf90, a tumor suppressor, can inhibit the occurrence and development of tumors. Therefore, we investigated the gene function of *C10orf90* in various tumors using multiple pan-cancer datasets. Pan-cancer analysis results reveal that the expression levels of *C10orf90* vary across different tumors and hold significant value in the clinical diagnosis and prognosis of patients with various tumors. In some cancers, the expression level of *C10orf90* is correlated with CNV, DNA methylation, immune subtypes, immune cell infiltration, and drug sensitivity in the tumors. In particular, in COAD, the *C10orf90* gene is implicated in multiple processes associated with COAD. Cell experiments demonstrate that *C10orf90* suppresses the proliferation and migration of colon cancer cells while promoting apoptosis. In summary, *C10orf90* plays a role in the onset and progression of various cancers and could potentially serve as an effective diagnostic and prognostic marker for cancer patients. Notably, in COAD, *C10orf90* inhibits the proliferation and migration of colon cancer cells, induces apoptosis, and is linked to the advancement of colon cancer.

## 1. Introduction

The mechanisms underlying the initiation of cancer remain poorly understood. Notably, the DNA damage response and defective repair processes may contribute to increased genomic instability, a critical factor in cancer development. With the vigorous development of genomics, researchers have discovered that specific DNA segments on chromosomes tend to exhibit a significant number of gaps and breaks in a highly non-random manner under replication stress. This phenomenon can lead to instability in chromosomal genes, designating these specific breakage regions as fragile sites [1].

*Chromosome 10 open reading frame 90* (*C10orf90*), also known as *D7Ertd443e* or *FATS*, is a fragile site gene located on chromosome 10q26.2, within the fragile region of chromosome FRA10F [2]. The *C10orf90* gene is rich in AT repeat sequences, with up to 56.8% AT base pairs. The cross-arrangement of double-stranded AT dinucleotide repeat sequences represents a fragile link, which is prone to gene breakage at this fragile site [3]. At the same time, multiple specialized DNA polymerases are required for the replicative synthesis of *C10orf90* to overcome differences in replication pausing caused by repetitive sequence elements or specific DNA sequence elements [4,5]; this requirement greatly increases the instability of the *C10orf90* gene. This explains the abnormal expression of *C10orf90* in various tumors, which is closely related to the rapid proliferation of cancer.

Fragile sites are characterized as “fragile”, areas prone to genomic instability, and these sites are hot spots for chromosomal rearrangements in the early stages of cancer development. Frequent gene deletions, amplifications, and rearrangements in cancer cells aggravate the replication pressure of DNA and increase the instability of fragile sites, leading to DNA breaks and accelerating the development of early-stage cancer [6,7]. Research results suggest that the N-terminal region of the C10orf90 protein can bind to the p53 protein, mediate p53 oligomerization to suppress the degradation of the p53 protein, and thereby enhance the cellular DNA damage repair ability [8]. In addition, the *C10orf90* protein can promote p21 acetylation, significantly decrease the activity of the proteasome subunit, attenuate the degradation rate of thep21 protein, and suppress cancer progression [9]. Some experimental evidence suggests that the *C10orf90* gene can upregulate *LC3I/II* and *ATG5* (markers of autophagy), increase the formation of mature autophagosomes and lysosomes, and promote cell apoptosis to inhibit NSCLC [10]. In addition, the deletion of C10orf90 can lead to mutations in *NRAS* (an oncogene) and *BRAF* (a proto-oncogene) [11], the latter two of which can confer greater therapeutic resistance and increased invasiveness to conjunctival melanoma [12,13]. Clinical data confirm a significant reduction in disease-free survival and a similar trend in 5-year overall survival for *C10orf90*-negative breast cancer patients compared to *C10orf90*-positive patients [14]. The *C10orf90* gene has been identified as having a significant inhibitory impact on the advancement of various tumors, including non-small cell lung cancer, conjunctival melanoma, and breast cancer. This observation indicates that the biological function of the *C10orf90* gene involves suppressing the proliferation of tumor cells. Previous studies have predominantly concentrated on exploring the genetic involvement of *C10orf90* in individual cancer types, without conducting a comprehensive pan-cancer analysis of *C10orf90*, especially in COAD. Consequently, the objective of this study is to analyze the pan-cancer functionality of the C10orf90 gene in order to enhance the understanding of its biological role in different malignant tumors.

In this research (Figure 1), we utilized the GEPIA2, HPA, TCGA, and GTEx databases to examine the transcriptional and translational statuses of C10orf90. Using bioinformatic methods, we performed prognostic and diagnostic value analysis of the C10orf90 gene, analysis of gene copy number variation, and DNA methylation, among others. Furthermore, we performed various cell experiments, such as CCK-8, plate cloning, Transwell, wound-healing, and apoptosis assays, to clarify the possible function of the *C10orf90* gene in inhibiting proliferation and migration in COAD cells. These experiments aimed to contribute additional data to elucidate the underlying biological mechanisms associated with the *C10orf90* gene.

## 2. Results

### 2.1. Expression and Subcellular Location of C10orf90

Utilizing the GEPIA2 and HPA databases, we performed a search for *C10orf90* mRNA and protein expression. C10orf90 protein expression was detected in various organs and tissues, including the testis, brain, breast, forearm, colon, kidney, liver, and pancreas (Figure 2A). The findings from the HPA database indicate that the C10orf90 protein is predominantly expressed in the testis, lung, placenta, and colon (Figure 2B). The GTEx and Consensus dataset databases demonstrate that *C10orf90* mRNA is mainly expressed in the spinal cord, brain, testis, salivary gland, breast, skin, adipose tissue, colon, and kidney (Figure 2C,D and Appendix A). Additionally, immunofluorescence localization of C10orf90 subcellular localization was obtained by staining the nucleus, microtubules, and endoplasmic reticulum in SK-MEL-30, U-2 OS, and U-251 MG cells. The results demonstrate that C10orf90 is primarily located in microtubules and cytoplasmic solutes (Figure 2E).

### 2.2. Expression of C10orf90 in Various Tumor Tissues

To assess the expression of *C10orf90* in multiple cancerous tissues, we conducted a joint analysis of *C10orf90* expression data from the TCGA and GTEx databases. This encompassed normal samples, tumor samples, and their corresponding adjacent tumor samples. A comparison of *C10orf90* expression levels in normal tissues with those in DLBC, LGG, LUAD, LUSC, PAAD, SKCM, THYM, UCS, and UCEC revealed a significant upregulation in all of these tumor types (*p* < 0.05). Conversely, *C10orf90* is downregulated in BRCA, COAD, GBM, HNSC, KICH, KIRC, KIRP, LIHC, OV, PRAD, READ, TGCT, and THCA (Figure 3A and Appendix A). The TCGA dataset analysis revealed varying levels of expression of *C10orf90* in tumor tissues and paired normal tissues of the following cancers: BRCA, CHOL, COAD, HNSC, KIRC, LUAD, LUSC, PRAD, and UCEC (Figure 3B and Appendix A). In comparison to adjacent normal tissues, *C10orf90* mRNA is significantly decreased in BRCA, COAD, HNSC, KIRC, and PRAD, while significantly increased in LUAD, LUSC, and UCEC (Appendix A). Similarly, analysis of HPA data indicates that the levels of the C10orf90 protein expression are decreased in liver and kidney cancer tissues compared to normal tissues (Figure 3C). Furthermore, we examined the transcript levels of *C10orf90* mRNA in different subtypes of various cancers, uncovering diverse expression patterns of *C10orf90* mRNA in different subtypes of BRCA, SKCM, and TGCT (Figure 3D and Appendix A).

### 2.3. Prognostic Relevance of C10orf90 Expression in Various Tumors

The clinical data sourced from TCGA were utilized to perform univariate Cox regression analysis, investigating the association between *C10orf90* expression and overall survival (OS), disease-specific survival (DSS), and progression-free interval (PFI) across diverse cancer types. As demonstrated in Figure 4A, elevated levels of *C10orf90* expression were significantly associated with worse overall survival rates among individuals diagnosed with BLCA, KICH, KIRC, LGG, STAD, UCEC, and UVM. Among these cancers, KIRC exhibited the most pronounced association with *C10orf90*. The forest plot and Cox assessment indicate that *C10orf90* is an unfavorable factor for disease-specific survival in patients with BLCA, ESCA, KICH, KIRC, LGG, LIHC, MESO, and UVM (Appendix A). In the analysis of cancer progression-free intervals, *C10orf90* was identified as a potential risk factor associated with BLCA, KICH, KIRC, LGG, LIHC, PRAD, STAD, UCEC, and UVM. In addition, there is a significant correlation between high levels of *C10orf90* expression and unfavorable prognostic outcomes in several types of cancer, including KIRC, LGG, OV, STAD, and UVM. Patients with high C10orf90 expression levels tended to have a worse prognosis compared to those with lower expression levels (Figure 4B–D). In conclusion, *C10orf90* expression demonstrated prognostic relevance in different types of tumors.

### 2.4. Diagnostic Value of C10orf90 in Various Cancers

As illustrated in Figure 5, we evaluated the diagnostic performance of *C10orf90* in various cancers using receiver operating characteristic (ROC) curves. Our findings indicate that *C10orf90* demonstrates favorable diagnostic value across multiple cancers, as evidenced by an area under the curve (AUC) exceeding 0.5 in 24 cancer types and even surpassing 0.7 in 13 types of cancers. *C10orf90* exhibits certain accuracy and high diagnostic value in predicting BRCA (AUC = 0.929), CESC (AUC = 0.709), COAD (AUC = 0.726), GBM (AUC = 0.856), HNSC (AUC = 0.710), KICH (AUC = 0.780) and KIRP (AUC = 0.708). The remaining cancers for which the model exhibited accuracy were KIRC (AUC = 0.748), KIRP (AUC = 0.708), LUAD (AUC = 0.701), LUSC (AUC = 0.826), PAAD (AUC = 0.714), PCPG (AUC = 0.704) and SARC (AUC = 0.861).

### 2.5. Association between C10orf90 Expression and CNV and Gene Methylation

We investigated the mechanisms behind gene copy number variation (CNV) and abnormal expression of *C10orf90* mRNA in tumors. The results are presented in Figure 6A and Appendix A. The expression of *C10orf90* in patients with TGCT, ESCA, LUSC, and HNSC was negatively correlated with CNV. However, in patients with GBM and SKCM, the expression of *C10orf90* mRNA was positively correlated with CNV expression, while the correlation was not significant in other tumor types. This suggests that CNV may not be the primary factor contributing to the abnormal expression of *C10orf90*. To address this issue, we assessed the promoter DNA methylation levels of *C10orf90* in various tumors. This investigation focused on the epigenetic regulation of *C10orf90* gene transcription (Figure 6B). The promoter of *C10orf90* exhibits hypomethylation in various cancers, such as BLCA, BRCA, CESC, CHOL, COAD, ESCA, HNSC, KIRC, KIRP, LIHC, LUAD, LUSC, PAAD, PCPG, READ, SARC, STAD, TCGT, THCA, and UCEC. Similarly, the correlation between *C10orf90* and genes related to N1-methyladenosine (m1A), 5-methylcytosine (m5C), and N6-methyladenosine(m6A) was assessed to explore the mechanisms underlying the inconsistent levels of *C10orf90* promoter DNA methylation observed in multiple cancers. The results demonstrated that *C10orf90* exhibited correlations with m1A, m5C, and m6A-related genes in most cancers, particularly in BLCA, HNSC, KICH, LUSC, OV, THCA, and UVM. In these cancers, *C10orf90* was positively correlated with the majority of methylation-related genes (Figure 6C and Appendix A). The aforementioned results indicate that the *C10orf90* gene may exert a functional influence by modulating CNV and gene methylation, thereby regulating gene expression levels in tumors.

### 2.6. Expression of C10orf90 in Different Immune and Molecular Subtypes of Various Cancers

We analyzed the levels of *C10orf90* expression across various cancer immune and molecular subtypes using the TISIDB database. The results, as illustrated in Figure 7A, demonstrate a notable disparity in the expression levels of *C10orf90* across various immune subtypes of multiple cancers, including BRCA, GBM, HNSC, LUSC, SARC, THCA, and UVM. For the molecular subtypes (Figure 7B), differential expression of *C10orf90* was observed among patients with BRCA, ESCA, HNSC, LGG, LIHC, LUSC, and SKCM. However, for other cancers, including immune subtypes such as BLCA, CHOL, ESCA, LGG, LIHC, SKCM, and USC, as well as molecular subtypes like GBM, the expression of C10orf90 is not statistically significant (Appendix A).

### 2.7. Correlation between C10orf90 Expression and Immune Infiltration

Given the presence of immune cell infiltration in the tumor microenvironment, which is closely linked to the efficacy of immunotherapy, we initially investigated the correlation between *C10orf90* and genes identified as immune checkpoint genes. The results demonstrated that among the immunostimulators, *C10orf90* exhibited a positive correlation with the immunostimulators in tumors such as COAD, PAAD, and UVM, and a negative correlation with SKCM, TGCT, and THCA cancers (Figure 8A and Appendix A). Conversely, *C10orf90* demonstrated a similar pattern of correlation with immunoinhibitors in tumors such as COAD, PAAD, and UVM, with a negative correlation observed in SKCM, TGCT, and THCA cancers (Figure 8B and Appendix A). Subsequently, the correlation between C10orf90 expression in pan-cancer and the degree of tumor-infiltrating immune cells was validated using the TIMER database. The correlation coefficients of six tumor-infiltrating immune cells (NK cells, monocytes, macrophages, endothelial cells, myeloid dendritic cells, and cancer-associated fibroblasts) were collected and presented in a heatmap format (Figure 8C). The most pronounced correlation between *C10orf90* expression levels and immune cell infiltration was observed in COAD, followed by BRCA and PAAD. Given the strongest association identified between *C10orf90* and immune infiltration in COAD, our analysis focused on exploring the relationship between *C10orf90* and 24 different subtypes of immune cells in COAD as a case study. The expression of the *C10orf90* gene was found to be positively correlated with the presence of macrophages, Tcm, Tem, T helper cells, neutrophils, pDC, and Th1 cells. Conversely, it is negatively correlated with the presence of NK CD56 bright cells and Th17 cells (Figure 8D,E). These findings indicate that *C10orf90* has the potential to influence the immune response by regulating genes related to immune modulation and the presence of immune infiltrating cells within tumors. This impact appears to be particularly significant in the context of COAD, as the extent of immune cell infiltration has been associated with the level of *C10orf90* expression.

### 2.8. PPI Network and Gene Function Enrichment Analysis of C10orf90

Previous research findings indicate that *C10orf90* expression is significantly downregulated in colon adenocarcinoma compared to normal and adjacent tissues, which has been demonstrated to have a high diagnostic value. In addition, the most significant correlation between the *C10orf90* gene and immune infiltration in COAD was utilized as a case study to explore the biological functions of *C10orf90* in COAD. This was achieved through protein–protein interaction (PPI) network analysis and gene function enrichment analysis of *C10orf90*. The PPI network analysis demonstrated that the C10orf90 protein may interact with associated proteins, including DOCK1, p53, CEP295, CALY, OR8B12, and SMIM19, with scores of 0.838, 0.568, 0.442, 0.418, 0.404 and 0.400, respectively (Figure 9A,B). The DOCK1 and p53 proteins exhibited a strong correlation with *C10orf90* in colon cancer patients. DOCK1 has been demonstrated to facilitate the movement and infiltration of cancer cells, while p53 proteins are crucial in inhibiting cancer cell growth. Considering the biological functions of DOCK1 and p53 proteins, it is reasonable to suggest that C10orf90 could also have a significant impact on COAD tumorigenesis and progression.

Gene enrichment analysis revealed that there were 3256 differential genes associated with *C10orf90* expression in COAD patients. Among these, 3218 were upregulated and 38 were down regulated (absLogFC > 1.5, *p*.adj < 0.01) (Figure 9C and Appendix A). Furthermore, the top 30 genes with the most significant positive correlation with the *C10orf90* gene were displayed in the gene expression heat map (Figure 9D). Concurrently, the disproportionately low number of downregulated genes prompted the analysis of 3218 upregulated genes that are positively associated with *C10orf90* gene expression for GO and KEGG pathway enrichment. At *p*.adj < 0.01, an analysis of GO enrichment revealed 11 biological processes (GO-BP), 13 cellular component (GO-CC), and 3 molecular function (GO-MF) pathways, in addition to 6 pathways identified by KEGG analysis (Appendix A). The bubble chart illustrated the top three enriched pathway information for BP, CC, and MF, as well as the top five pathways for KEGG (Figure 9E,F). The most significant findings of the GO analyses were the bitter taste receptor activity of MF (GO:0033038), the nucleosome of CC (GO:0000786), the quadruple SL/U4/U5/U6 snRNP of BP (GO:0000353), and the KEGG analysis was most notable for systemic lupus erythematosus (hsa05322). The gene enrichment analysis suggests that in COAD, the *C10orf90* gene may influence tumor progression through the above-mentioned pathway and thus may be involved in the development of COAD.

### 2.9. Sensitivity Analysis of C10orf90 Related Drugs

We investigated the drug sensitivity of *C10orf90* expression in tumors. The CTPR database showed a correlation between *C10orf90* expression and drug sensitivity. Several drugs positively correlate with *C10orf90* expression, the top three being afatinib (an ErbB family blocker), ibrutinib (a Bruton’s tyrosine kinase inhibitor), and austocystin D (a DNA damage agent). In contrast, the 50% inhibitory concentration (IC50) values of the drugs dabrafenib (a BRAF inhibitor) and vemurafenib (a BRAF inhibitor) were negatively associated with *C10orf90* expression. The inhibitory concentration (IC50) values were negatively correlated with *C10orf90* expression (Figure 10A and Appendix A). Additionally, based on the GDSC drug sensitivity results, the top three drugs positively correlated with *C10orf90* expression were EKB-569 (an EGFR-TK inhibitor), AICAR (an AMPK activator) and PHA-793887 (a CDK1 inhibitor). Conversely, the top three drugs negatively correlated with *C10orf90* expression were PLX4720 (a ZAK inhibitor), dabrafenib, and SB590885 (a BRAF inhibitor) (Figure 10B and Appendix A).

### 2.10. Overexpression of C10orf90 Inhibits Colon Cancer Cell Proliferation and Tumor Migration

Given that the *C10orf90* gene is expressed at low levels in COAD, we utilized lentiviral constructs of two murine-derived colon cancer cell lines (CT26 and MC38) overexpressing the *D7Ertd443e* (*C10orf90* homolog) gene to investigate the potential role of the *D7Ertd443e* gene in colon cancer. The results demonstrated the efficacy of the protocol for constructing CT26 and MC38 cells overexpressing the *D7Ertd443e* gene (Figure 11A). Subsequent to lentiviral infection, the results from CCK-8 and plate cloning assays demonstrated that the overexpression of *D7Ertd443e* inhibited COAD cell proliferation (Figure 11B,C). Furthermore, the Transwell and wound-healing assays revealed that overexpression of *D7Ertd443e* inhibited the migratory potential of the identified COAD cells (Figure 11D,E). It is noteworthy that overexpression of *D7Ertd443e* was found to induce apoptosis in COAD cells (Figure 11F). In conclusion, these findings indicate that *D7Ertd443e* (*C10orf90* homolog) has the potential to significantly inhibit colon cancer cell proliferation and tumor migration while promoting apoptosis.

## 3. Discussion

The *C10orf90* gene is characterized by a distinctive fragile site structure, which is susceptible to disruption during the rapid proliferation of tumor cells. This can lead to genomic instability and impede the progression of cancer. Studies have demonstrated that the C10orf90 protein can inhibit the degradation rate of p53 and p21 proteins, thereby contributing to the occurrence and development of cancers, including non-small cell lung cancer, breast cancer, conjunctival melanoma, and pterygium and pinguecula. The actions of this protein have been shown to influence the survival outcomes and prognoses of individuals with these types of cancer. Therefore, we conducted a pan-cancer analysis of the *C10orf90* gene using multiple databases and bioinformatics tools. Subsequently, we combined this analysis with cellular experiments to explore the potential role of C10orf90 in COAD.

The findings of the pan-cancer study suggest that the C10orf90 protein is present in a variety of bodily organs and tissues, including the testis, brain, breast, forearm, colon, kidney, liver, and pancreas. Subcellular localization suggests that C10orf90 functions in microtubules and cytoplasmic solutes. In comparison to normal tissues, *C10orf90* is significantly upregulated in DLBC, LGG, LUAD, LUSC, PAAD, SKCM, THYM, UCS, and UCEC. Conversely, *C10orf90* is downregulated in BRCA, COAD, GBM, HNSC, KICH, KIRC, KIRP, LIHC, OV, PRAD, READ, TGCT and THCA. In comparison to adjacent tissues, *C10orf90* mRNA is significantly downregulated in BRCA, COAD, HNSC, KIRC, and PRAD, and significantly upregulated in LUAD, LUSC, and UCEC. Moreover, the protein expression level of C10orf90 is decreased in liver cancer and kidney cancer tissues, and the expression varies among different subtypes of BRCA, SKCM, and TGCT. These findings indicate that *C10orf90* expression levels are heterogeneous across multiple tumors, suggesting the need for further investigation and analysis.

Our results indicate that *C10orf90* has prognostic and diagnostic value in various cancers. Univariate Cox regression and Kaplan–Meier analysis demonstrated that *C10orf90* expression was significantly associated with overall survival, disease-specific survival, and progression-free interval in KIRC, LGG, OV, STAD, and UVM. Patients with high *C10orf90* expression exhibited a worse prognosis than those exhibiting low *C10orf90* expression. However, it is important to note that the expression level of *C10orf90* is generally downregulated in patients with tumors. While the low expression of the *C10orf90* gene may not be a primary factor influencing survival rates, it still holds some prognostic significance. In terms of diagnostic value, the area under the ROC curve is above 0.5 in 24 types of cancer and exceeds 0.7 in 13 types of cancer, with BRCA having the highest AUC (0.929). It has been demonstrated that the expression of *C10orf90* is downregulated in breast cancer samples and that cancer patients exhibit significant genetic variations in the *C10orf90* gene, which are associated with susceptibility to breast cancer [15]. These findings indicate that *C10orf90* has superior prognostic prediction and diagnostic value in these cancers, with potential applications in the clinical setting.

Tumor tissues exhibit genetic copy number variation and abnormal DNA methylation, which regulate cancer development [16,17]. Therefore, we analyzed the relationship between *C10orf90* and CNV and DNA methylation using the GSCA and UALCAN databases. We observed a correlation between *C10orf90* and CNV in 6 cancer types. Additionally, the promoter of *C10orf90* exhibited hypomethylation in 20 cancer types. Furthermore, *C10orf90* showed a correlation with m1A, m5C, and m6A-related genes in tumors. In BLCA, HNSC, KICH, LUSC, OV, THCA, and UVM, C10orf90 was positively correlated with the majority of methylation-related genes. DNA methylation is a common biochemical process that regulates gene expression and gene stability. However, in most tumor cells, aberrant methylation is prevalent in the promoters of some crucial tumor suppressor genes, leading to the dysregulation of DNA repair and chromosome stability genes. This, in turn, results in uncontrolled cancer development [18]. In conclusion, the results indicate that *C10orf90* may exert gene functions in tumors by affecting gene copy number variation and DNA methylation to regulate gene expression levels.

The capacity of immune cells to eliminate tumor cells and impede cancer progression is a crucial aspect of cancer immunotherapy. Consequently, investigating the extent of immune cell infiltration within the tumor microenvironment is essential for advancing our understanding of this therapeutic approach [19]. Studies have demonstrated that *C10orf90* can impede the TLR4-NF-κB activation pathway, inhibit the degradation of IκBα to suppress the NF-κB activation pathway and stimulate enhanced proliferation and activation of CD4^+^ T cells. The *C10orf90* gene has been shown to promote the phenotypic shift from M2-like macrophages to anti-tumor M1-like macrophages. This ultimately inhibits melanoma and pancreatic tumorigenesis and development. This suggests that *C10orf90* plays an important role in immune infiltration [20]. Consequently, we analyzed the expression levels of *C10orf90* in immune subtypes and molecular subtypes of multiple cancers. The expression levels of *C10orf90* are significantly different in the immune subtypes of tumors such as BRCA, GBM, HNSC, LUSC, SARC, THCA, and UVM. For molecular subtypes, *C10orf90* exhibits different expression patterns in patients with BRCA, ESCA, HNSC, LGG, LIHC, LUSC, and SKCM. Immune checkpoint genes are mainly categorized into two groups: immune stimulation and immune inhibition. The data indicate that among immunostimulators, *C10orf90* is positively correlated with immunostimulators in tumors such as COAD, PAAD, and UVM, while it is negatively correlated in cancers such as SKCM, TGCT, and THCA. In addition, *C10orf90* is positively correlated with immunoinhibitors in tumors such as COAD, PAAD, and UVM, while it is negatively correlated in cancers such as SKCM, TGCT, and THCA. In the TIMER database, we identified a correlation between *C10orf90* expression and six types of tumor-infiltrating immune cells, including NK cells, monocytes, macrophages, endothelial cells, myeloid dendritic cells, and cancer-associated fibroblasts. Interestingly, the *C10orf90* gene shows the strongest association with immune infiltration in COAD. Further analysis revealed a strong positive correlation between *C10orf90* gene expression and macrophages, Tcm, Tem, T helper cells, neutrophils, pDC, and Th1 cells in COAD. Conversely, a strong negative correlation was observed between NK CD56 bright cells and Th17 cells. According to reports, during the pathogenesis of COAD, M2-like macrophages secrete IL-1β, which induces Wnt signaling and supports tumor cell growth [21]. T helper cells are a crucial component of the adaptive immune system. Th1 cells can enhance the effector function of tumor-infiltrating lymphocytes and exert anti-tumor killing function, as evidenced by research in reference [22]. Th17 cells secrete IL-17A, which induces pyroptosis in colon cancer cells by stimulating the production of reactive oxygen species (ROS). Additionally, IL-17A recruits more CD8^+^ T cells to the tumor microenvironment, thereby demonstrating an anti-tumor immune response [23]. These findings suggest that *C10orf90* might modulate tumor immunity through its influence on immune regulatory genes and immune infiltrating cells, especially in COAD. This highlights the importance of studying *C10orf90* expression as part of tumor immunotherapy.

The PPI network was constructed using STRING, and the C10orf90 protein may be involved in interactions with related proteins, including DOCK1, p53, CEP295, CALY, OR8B12, and SMIM19. DOCK1 functions as a GTPase exchange factor that activates Rac proteins and stimulates actin polymerization at the membrane surface, thereby altering the cytoskeletal structure and cellular morphology to enhance the migration and invasive capabilities of tumor cells such as glioma, breast cancer, ovarian cancer, and other types of tumors [24]. The *TP53* gene functions as a tumor suppressor by primarily promoting cellular apoptosis and facilitating the repair of DNA damage [25,26]. CEP295 is an essential protein for centriole formation and interacts directly with microtubules through its distinctive structural domains. It recruits upstream effectors in mitotic S and G2 phases, assembles centriole microtubules, and participates in post-translational modifications [27]. CALY is primarily involved in the clathrin-mediated endocytic machinery and regulates vesicular trafficking [28]. Although DOCK1, p53, CEP295, and CALY were identified as proteins interacting with C10orf90, further experimental validation of their interactions during carcinogenesis is necessary. In addition, we performed GO and KEGG pathway enrichment analyses. The differential genes linked to *C10orf90* expression in COAD patients were subjected to analysis. The GO results showed that *C10orf90* may regulate tumorigenesis and development through bitter taste receptor activity, taste receptor activity, nucleosome, DNA packaging complex, formation of quadruple SL/U4/U5/U6 snRNP and mRNA trans-splicing via spliceosome pathway effects. The KEGG analysis revealed that *C10orf90* was associated with systemic lupus erythematosus, alcoholism, neutrophil extracellular trap formation, taste transduction, olfactory transduction, and neuroactive ligand-receptor interaction. These results indicate that in COAD, the *C10orf90* gene plays important biological functions through multiple signaling pathways.

The relationship between *C10orf90* and drug sensitivity was investigated using data from the CTPR and GDSC databases. The CTPR and GDSC databases identified 65 and 70 drugs, respectively, which may be associated with the *C10orf90* gene. Some of these drugs include afatinib, ibrutinib, austocystin D, dabrafenib, vemurafenib, EKB-569, AICAR, PHA-793887, PLX4720, and SB590885. These drugs are used clinically to prevent cancer progression, with afatinib [29], ibrutinib [30], dabrafenib [31], vemurafenib [32], AICAR [33], and PLX4720 [34] being commonly used anti-cancer drugs for COAD patients. The *C10orf90* gene may serve as a marker for predicting the therapeutic efficacy of afatinib, ibrutinib, dabrafenib, vemurafenib, AICAR, and PLX4720 in patients with COAD.

Finally, the bioinformatics results were used to confirm that overexpression of the *C10orf90* gene could inhibit the proliferation and migration of colon cancer cells, induce apoptosis, and play a positive anticancer role at the cellular level. This was achieved through the utilization of the following assays: CCK-8, plate cloning, Transwell, wound healing, and apoptosis assays.

This study is subject to certain limitations. Despite utilizing multiple databases and bioinformatics tools to analyze the role of the *C10orf90* gene in various cancers, there is still a lack of extensive clinical samples and experimental data for validation. In addition, we have preliminarily demonstrated that at the cellular level, high expression of *C10orf90* could inhibit the proliferation and migration of colon cancer cells and induce apoptosis. Therefore, further validation in vivo through animal experiments is necessary, as well as excavating the deep molecular mechanisms to reveal the biological effects of the *C10orf90* gene in COAD.

## 4. Materials and Methods

### 4.1. Investigation of the Expression Patterns and Subcellular Localization of C10orf90

We conducted a comprehensive search of the Gene Expression Profiling Interactive Analysis 2 (http://gepia2.cancer-pku.cn/#general/ accessed on 22 May 2024), the Human Protein Atlas (https://www.proteinatlas.org/ accessed on 22 May 2024), the Cancer Genome Atlas (https://cancergenome.nih.gov/ accessed on 22 May 2024), and the Genotype-Tissue Expression (https://gtexportal.org/ accessed on 22 May 2024) databases to retrieve protein and RNA expression profiles of C10orf90 from normal samples [35,36,37]. Then, we used log2 transformation to process the RNAseq data from TCGA and GTEx. We downloaded the subcellular localization of C10orf90 in human cancer cell lines (SK-MEL-30, U-2 OS, and U-251 MG) from the HPA database. In this context, green represents the C10orf90 protein, blue indicates the nucleus, red denotes microtubule proteins, and yellow signifies the endoplasmic reticulum. The 33 cancer types included adrenocortical carcinoma (ACC), bladder urothelial carcinoma (BLCA), breast invasive carcinoma (BRCA), cervical squamous cell carcinoma and endocervical adenocarcinoma (CESC), cholangiocarcinoma (CHOL), colon adenocarcinoma (COAD), lymphoid neoplasm diffuse large B-cell lymphoma (DLBC), esophageal carcinoma (ESCA), glioblastoma multiforme (GBM), head and neck squamous cell carcinoma (HNSC), kidney chromophobe (KICH), kidney renal clear cell carcinoma (KIRC), kidney renal papillary cell carcinoma (KIRP), acute myeloid leukemia (LAML), Brain lower grade glioma (LGG), liver hepatocellular carcinoma (LIHC), lung adenocarcinoma (LUAD), lung squamous cell carcinoma (LUSC), mesothelioma (MESO), ovarian serous cystadenocarcinoma (OV), pancreatic adenocarcinoma (PAAD), pheochromocytoma and paraganglioma (PCPG), prostate adenocarcinoma (PRAD), rectum adenocarcinoma (READ), sarcoma (SARC), skin cutaneous melanoma (SKCM), stomach adenocarcinoma (STAD), testicular germ cell tumors (TGCT), thyroid carcinoma (THCA), thymoma (THYM), uterine corpus endometrial carcinoma (UCEC), uterine carcinosarcoma (UCS), and uveal melanoma (UVM).

### 4.2. Analysis of Prognostic and Diagnostic Value of C10orf90

In the clinical relevance section of the Xiantao tool (https://www.xiantao.love/ accessed on 29 May 2024), patients underwent analysis utilizing Cox regression and the Kaplan–Meier method to assess prognostic factors such as overall survival, disease-specific survival, and progression-free interval. Statistical analysis was conducted using the Wilcoxon test, with *p* < 0.05 indicating a significant difference. The expression data of the C10orf90 gene were employed in the construction of the ROC curve, with the area under the curve serving as a metric to assess its accuracy.

### 4.3. Analysis of CNV and Gene Methylation Correlations

The “Mutation” component of the Gene Set Cancer Analysis database (http://bioinfo.life.hust.edu.cn/GSCA/ accessed on 29 May 2024) was employed to examine the association between *C10orf90* and gene copy number variations across different types of tumors [38]. We utilized the University of Alabama at Birmingham Cancer Database (http://ualcan.path.uab.edu/ accessed on 29 May 2024) to investigate the DNA methylation status of the *C10orf90* promoter [39]. The β value was employed to quantify the level of DNA methylation, with values indicating low methylation (β: 0.25–0.3) and high methylation (β: 0.5–0.7). The associations between *C10orf90* and genes associated with m1A, m5C, and m6A modifications were assessed through a heatmap analysis.

### 4.4. Expression of C10orf90 across Various Immune and Molecular Subtypes of Cancer

We utilized the “Subtype” module within the TISIDB database (http://cis.hku.hk/TISIDB/ accessed on 2 June 2024) to investigate the expression patterns of *C10orf90* across different immune and molecular subtypes present in various tumor types [40], such as C1 (wound healing), C2 (IFN-γ dominant), C3 (inflammatory), C4 (lymphocyte depleted), C5 (immune quiescent), and C6 (TGF-β dominant).

### 4.5. Relationship between C10orf90 Expression and Immune Infiltration

The XCELL algorithm, derived from the Tumor Immune Estimation Resource 2.0 database, was used to evaluate the impact of *C10orf90* on immune cell infiltration within tumors [41]. At the same time, the Xiantao tool was used to investigate the correlation between *C10orf90* in COAD patients and various immune cell subtypes, as well as its association with specific immune cell subpopulations.

### 4.6. Protein–Protein Interaction Network Analysis and Enrichment Analysis of Gene Functions

The Search Tool for the Retrieval of Interacting Genes/Proteins database (www.string-db.org/ accessed on 6 June 2024) was used for all protein–protein interaction data [42]. Based on the TCGA data, the genes COAD and *C10orf90* were found to be co-expressed (with absLogFC > 1.5 and *p* < 0.01), followed by enrichment analyses using GO and KEGG databases.

### 4.7. Analysis of Drug Sensitivity in Association with C10orf90

Drug sensitivity of *C10orf90* gene expression was analyzed using the Genomics of Drug Sensitivity in Cancer (https://www.cancerrxgene.org/ accessed on 6 June 2024) and the Cancer Therapeutics Response Portal (http://portals.broadinstitute.org/ctrp/ accessed on 6 June 2024) databases [43,44].

### 4.8. Cell Lines, Culture and Transfection

The mouse colon cancer cell lines CT26 and MC38 were obtained from the Cell Resource Center, IBMS, CAMS/PUMC (Beijing, China). The cell culture environment was set at 37 °C, 5% CO_2,_ in appropriate humidity. Specific cell culture parameters included the use of 10% fetal bovine serum (FBS) in DMEM basal medium purchased from Pricella, Wuhan, China. Recombinant lentivirus (OE-*D7Ertd443e*) for *D7Ertd443e* overexpression and lentiviral vector (OE-NC) for negative control were purchased and infected into CT26 and MC38 cells. Moreover, 10 μg/mL of puromycin was incubated for 48 h to screen the infected cells.

### 4.9. QRT-PCR

RNA was extracted using the TRIzol method (TaKaRa, Kusatsu, Japan), reverse transcribed into cDNA, and used in a qRT-PCR kit (Vazyme, Nanjing, China) for detection. PCR detection included reactions at 95 °C (30 s), 95 °C (10 s), and 60 °C (15 s) for 40 cycles and 72 °C (30 s). The sequences for the mouse *D7Ertd443e* primers were (F) 5′-AGACCATTAAACACTTCCCTCCT-3′ and (R) 5′-AAGGGCAATAAAACCCAGGCA-3′.

### 4.10. CCK-8 Assay

CT26 and MC38 cells infected with OE-D7Ertd443e and OE-NC were inoculated with 2 × 10^3^ cells in a 96-well plate, and 100 μL/well was cultured in the culture system for 24 h, 48 h, 72 h, and 96 h, respectively, and added to 10 μL of CCK-8 reagent (GLPBIO, Montclair, NY, USA); the OD values were measured after 1 h.

### 4.11. Plate Cloning Assay

CT26 and MC38 cells infected with OE-*D7Ertd443e* and OE-NC were inoculated with 2 × 10^3^ cells in 6-well plates and cultured for 10 days. The cells were rinsed with PBS buffer, treated with methanol for 30 min for fixation, and subsequently stained with 0.1% crystal violet solution for 30 min. They were then washed with pure water, dried, and photographed.

### 4.12. Transwell Assay

CT26 and MC38 cell lines exposed to OE-*D7Ertd443e* and OE-NC were inoculated with 1 × 10^3^ cells in the upper chamber of a Transwell kit (Corning, New York, NY, USA). The upper compartment was supplied with serum-free culture medium, while the lower compartment was supplied with DMEM culture medium supplemented with 20% FBS. The cells were rinsed with PBS buffer, treated with methanol for 30 min for fixation, and subsequently stained with 0.1% crystal violet solution for 30 min. They were then washed with pure water, dried, and photographed.

### 4.13. Wound-Healing Assay

CT26 and MC38 cells infected with OE-*D7Ertd443e* and OE-NC were inoculated with 1 × 10^6^ cells in 6-well plates and cultured overnight. Parallel scratches were made on the cell layer with a 200 μL lance tip, washed three times with PBS buffer, and photographed at 0 h, 24 h, 48 h, and 72 h to observe changes in scratch width.

### 4.14. Apoptosis Assay

CT26 and MC38 cell lines infected with OE-*D7Ertd443e* and OE-NC were seeded in a 6-well plate at a density of 1 × 10^6^ cells and then incubated for 24 h. The cells were then collected in flow-through tubes and washed with PBS buffer; they were stained with 5 μL of Annexin V-APC and 5 μL of 7-AAD (Beyotime, Shanghai, China) and detected by a machine.

### 4.15. Statistical Analysis

Biological information obtained in this study was analyzed in databases. All cell experiment data were statistically analyzed using GraphPad Prism 9.0. A t-test was employed for comparisons between two groups, while ANOVA was utilized for comparisons among multiple groups. Data are presented as the mean ± standard error of the mean (Mean ± SEM). *p* < 0.05 is considered statistically significant. (* *p* < 0.05, ** *p* < 0.01, *** *p* < 0.001).

## 5. Conclusions

*C10orf90*, a gene known for its tendency to be expressed at low levels in specific tissues, exhibits differential expression levels in different tumors. This gene has been demonstrated to possess certain diagnostic and prognostic values. Moreover, the expression of *C10orf90* is correlated with several other factors, including CNV, DNA methylation, immune checkpoint genes, immune cell infiltration, and drug sensitivity in tumors. Specifically, in COAD, the *C10orf90* gene is involved in several immune-related processes and can inhibit the proliferation and migration of colorectal cancer cells. Consequently, the *C10orf90* gene shows promise as a diagnostic and prognostic biomarker in tumors, opening up new avenues for further research.

## Figures and Tables

**Figure 1 ijms-25-10496-f001:**
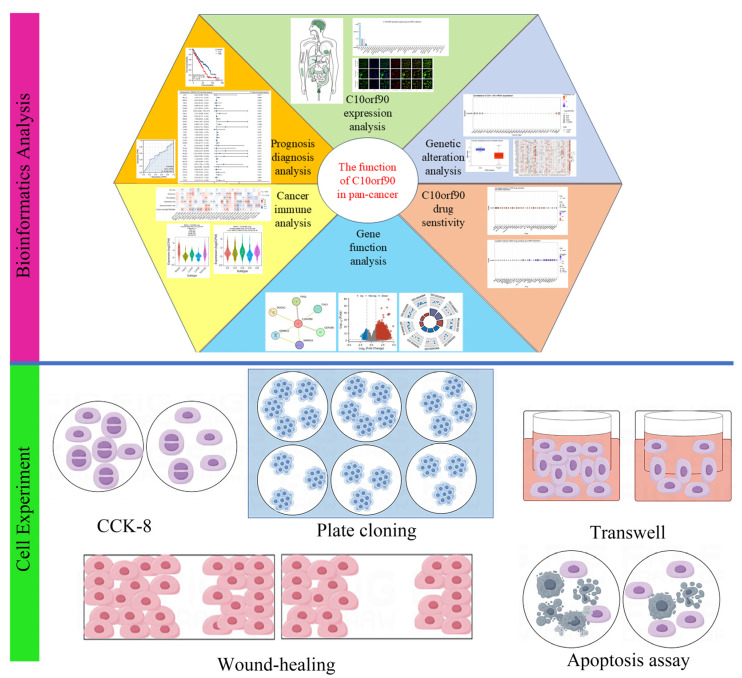
Main workflow of this study.

**Figure 2 ijms-25-10496-f002:**
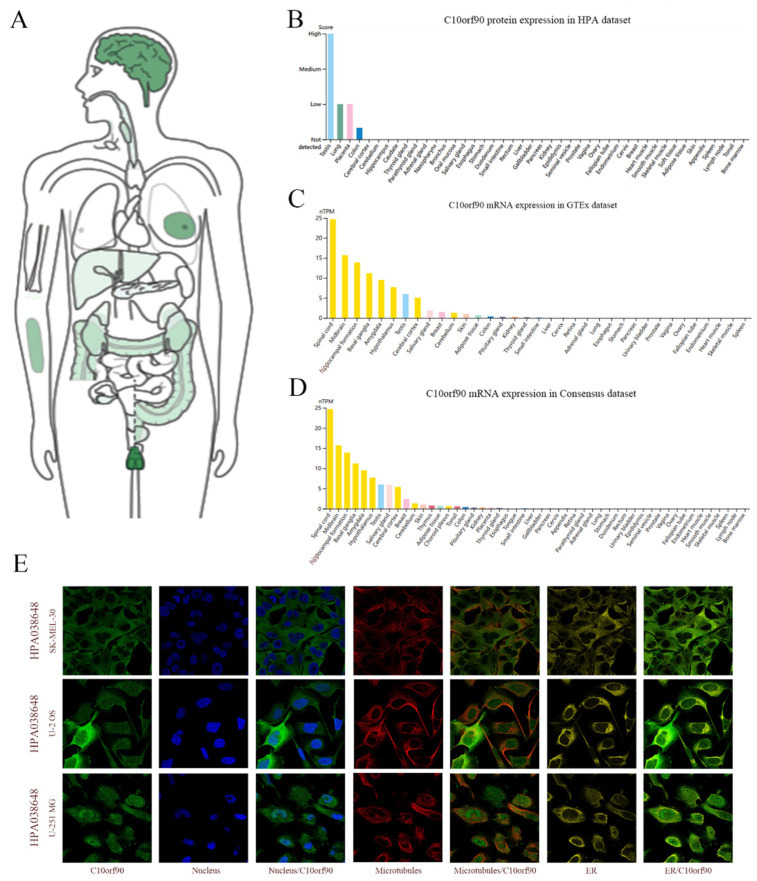
Expression and subcellular location of C10orf90. (**A**) Expression of C10orf90 protein in normal human tissues in GEPIA2 database. (**B**) Expression of C10orf90 protein in HPA database (Different colors represent different tissues, and the same applies below). (**C**) Expression of *C10orf90* mRNA in GTEx database. (**D**) Expression of *C10orf90* mRNA in Consensus dataset database. (**E**) Immunofluorescence staining of the subcellular localization of C10orf90.

**Figure 3 ijms-25-10496-f003:**
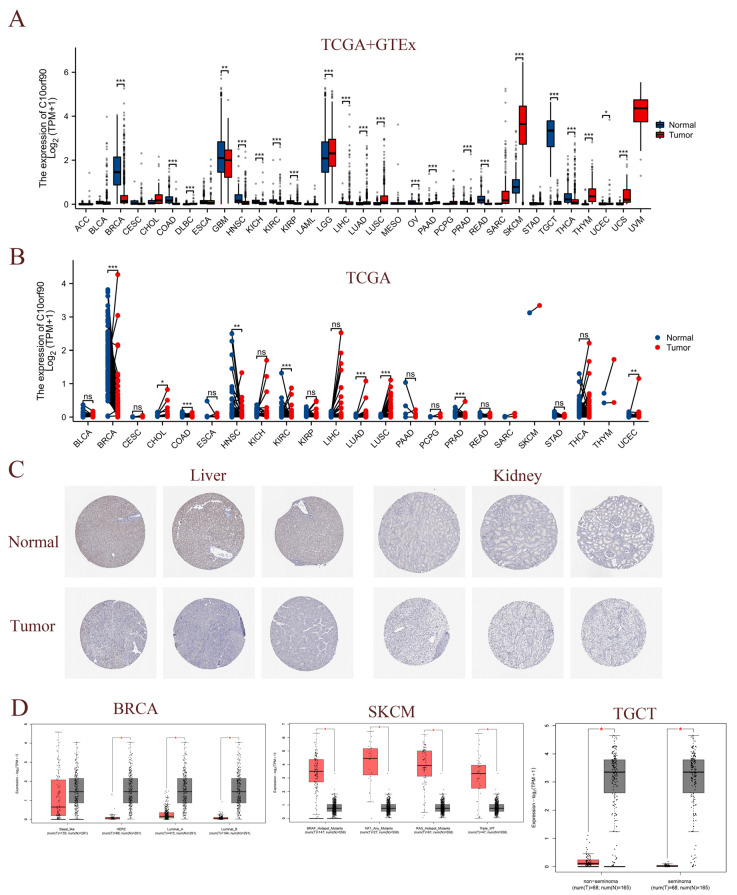
Expression of *C10orf90* in various tumor tissues. (**A**) Expression of *C10orf90* mRNA in TCGA + GTEx database. (**B**) Expression of *C10orf90* mRNA in TCGA database. (**C**) In HPA data, the expression of C10orf90 protein is lower in liver cancer and kidney cancer compared to normal tissues (antibody HPA075229). (**D**) GEPIA2 database analysis of *C10orf90* expression in different subtypes of BRCA, SKCM, and TGCT. (* *p* < 0.05, ** *p* < 0.01, *** *p* < 0.001).

**Figure 4 ijms-25-10496-f004:**
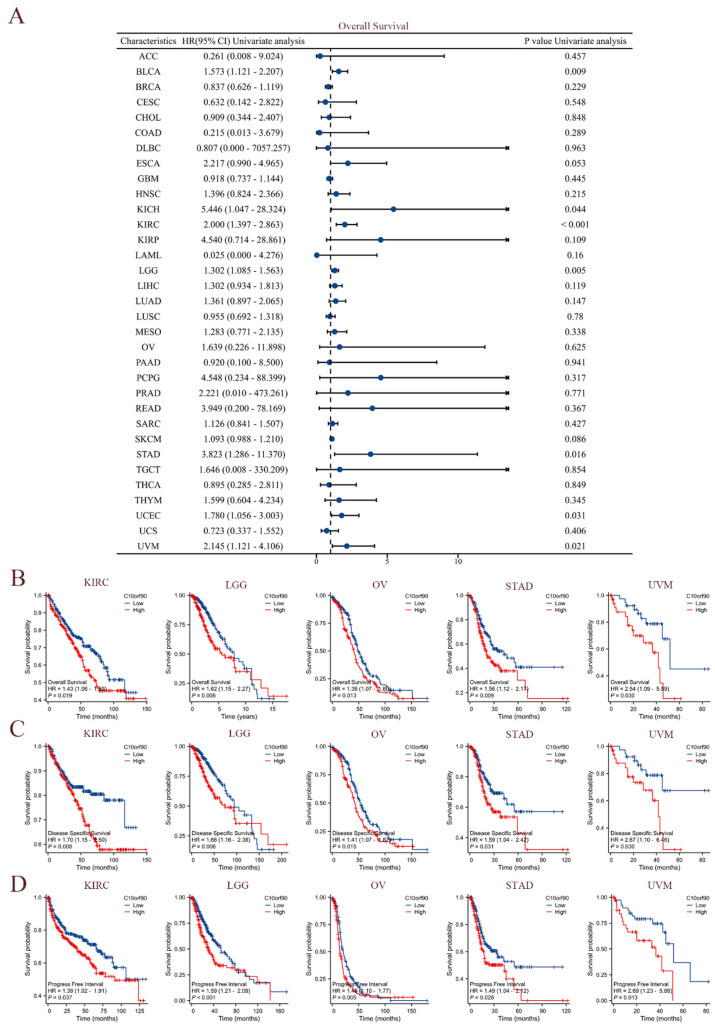
Prognostic correlation of *C10orf90* expression in multiple tumors. (**A**) The expression *C10orf90* is associated with overall survival. (**B**) Overall survival K-M curves of *C10orf90* expression in KIRC, LGG, OV, STAD, and UVM patients. (**C**) Disease-specific survival K-M curves of *C10orf90* expression in KIRC, LGG, OV, STAD, and UVM patients. (**D**) Progression-free interval K-M curves of *C10orf90* expression in KIRC, LGG, OV, STAD, and UVM patients.

**Figure 5 ijms-25-10496-f005:**
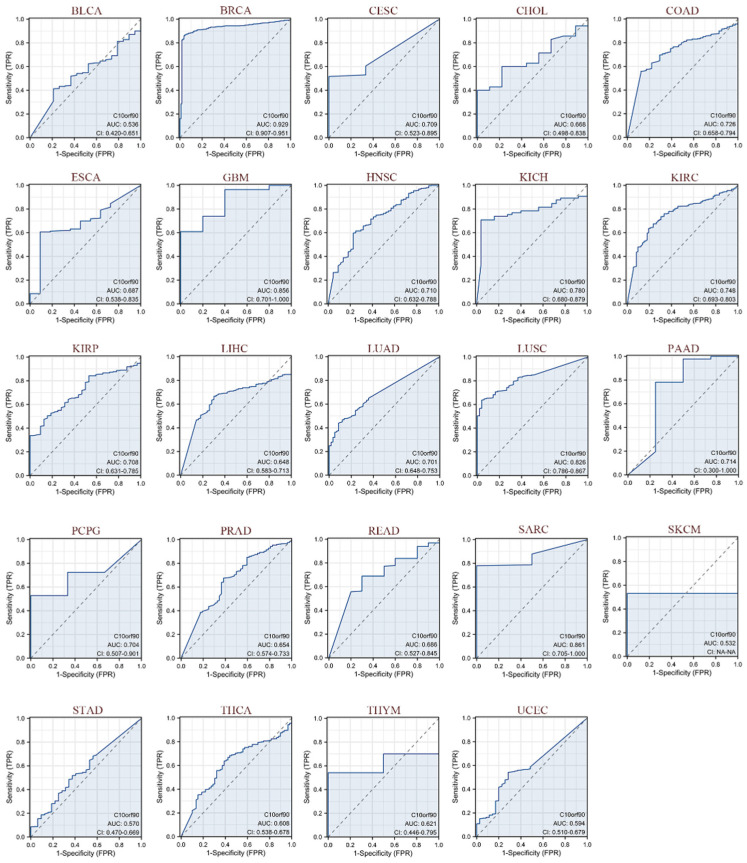
ROC curves of C10orf90 in multiple cancers. Cancers with AUC > 0.5 include BLCA, BRCA, CESC, CHOL, COAD, ESCA, GBM, HNSC, KICH, KIRC, KIRP, LIHC, LUAD, LUSC, PAAD, PCPG, PRAD, READ, SARC, SKCM, STAD, THCA, THYM, and UCEC. (Blue curve is the ROC curve and the dotted line is the baseline).

**Figure 6 ijms-25-10496-f006:**
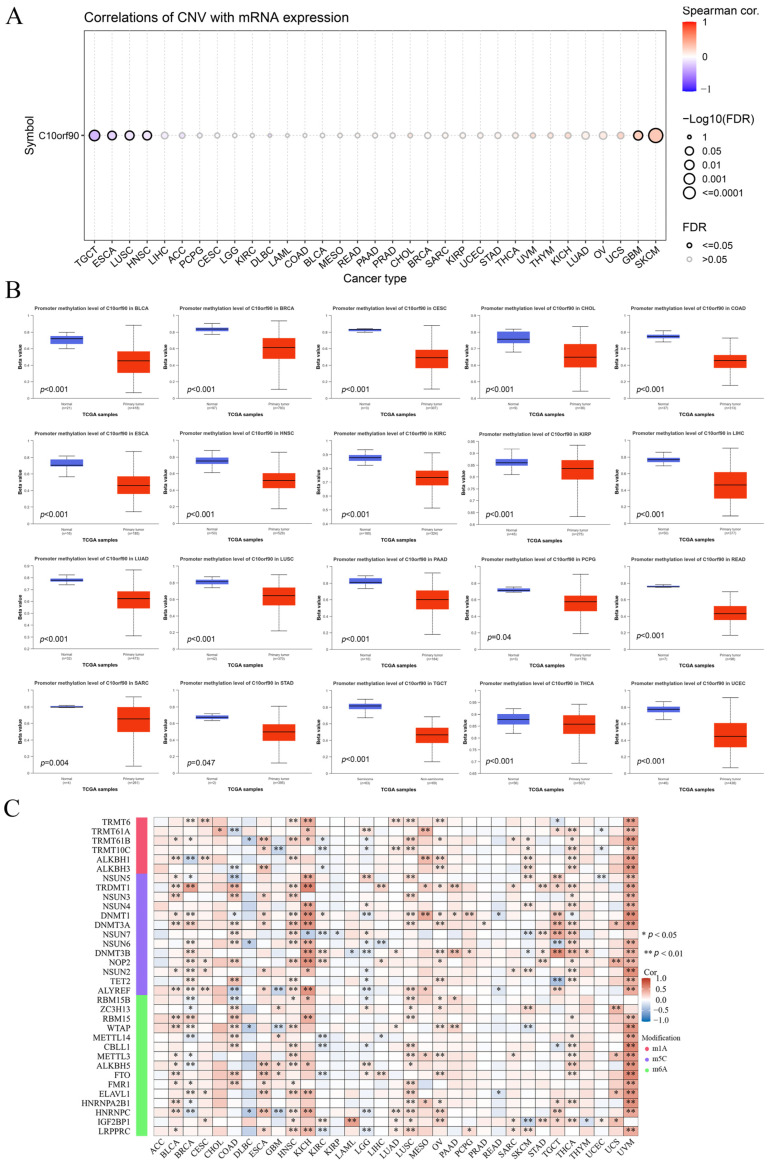
Association between *C10orf90* expression and CNV and gene methylation. (**A**) Correlation between CNV and C10orf90 expression in GSCA database. (**B**) Promoter methylation levels of *C10orf90* in 20 cancers from UALCAN. (**C**) Correlation analysis of *C10orf90* expression with mRNA modification methylation regulators.

**Figure 7 ijms-25-10496-f007:**
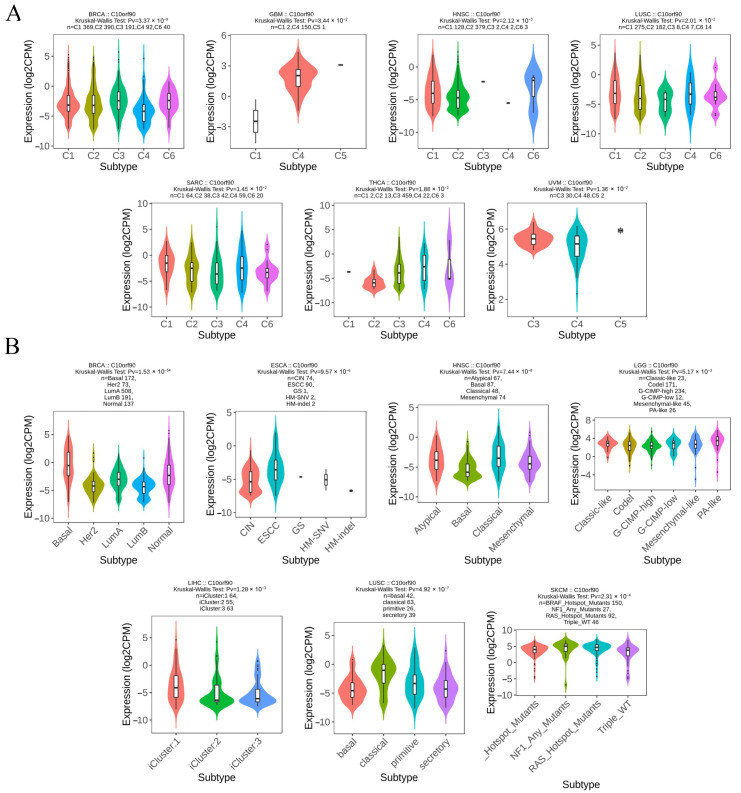
Expression of *C10orf90* in different immune and molecular subtypes of various cancers. (**A**) Correlation between *C10orf90* and immune subtypes of cancers, including BRCA, GBM, HNSC, LUSC, SARC, THCA, and UVM. (**B**) Correlation between *C10orf90* and cancer molecular subtypes, including BLCA, CHOL, ESCA, LGG, LIHC, SKCM, and USC.

**Figure 8 ijms-25-10496-f008:**
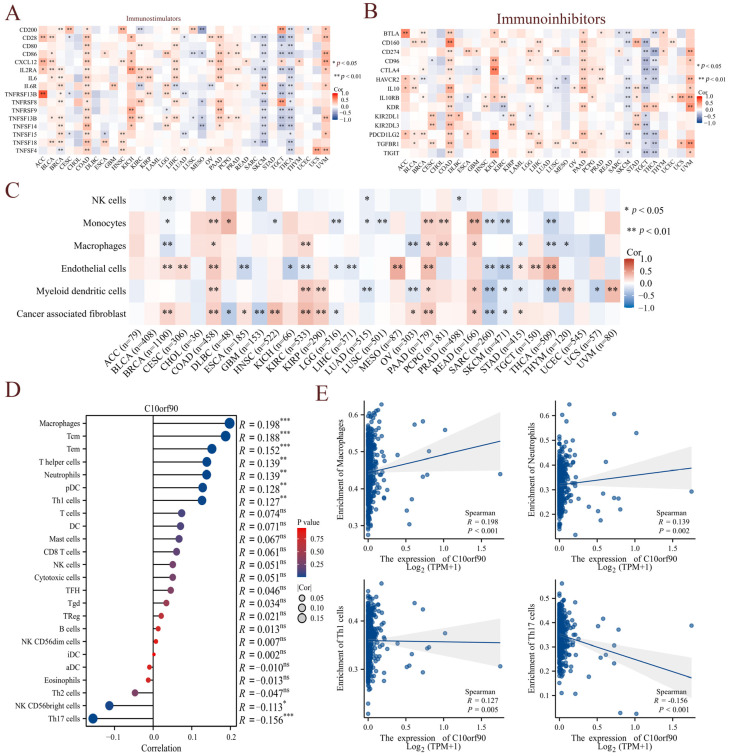
Correlation between *C10orf90* expression and immune infiltration. (**A**) Correlation between *C10orf90* and immunostimulators. (**B**) Correlation between *C10orf90* and immunoinhibitors. (**C**) Correlation between *C10orf90* and immune cell infiltration. (**D**) Correlation between *C10orf90* and immune cell subtypes in COAD. (**E**) Correlation between *C10orf90* and macrophages, neutrophils, Th1 cells, and Th17 cells in COAD. (Blue points represent the expression levels of C10orf90 in different samples, and the shadow indicate the confidence intervals of the data) (* *p* < 0.05, ** *p* < 0.01, *** *p* < 0.001).

**Figure 9 ijms-25-10496-f009:**
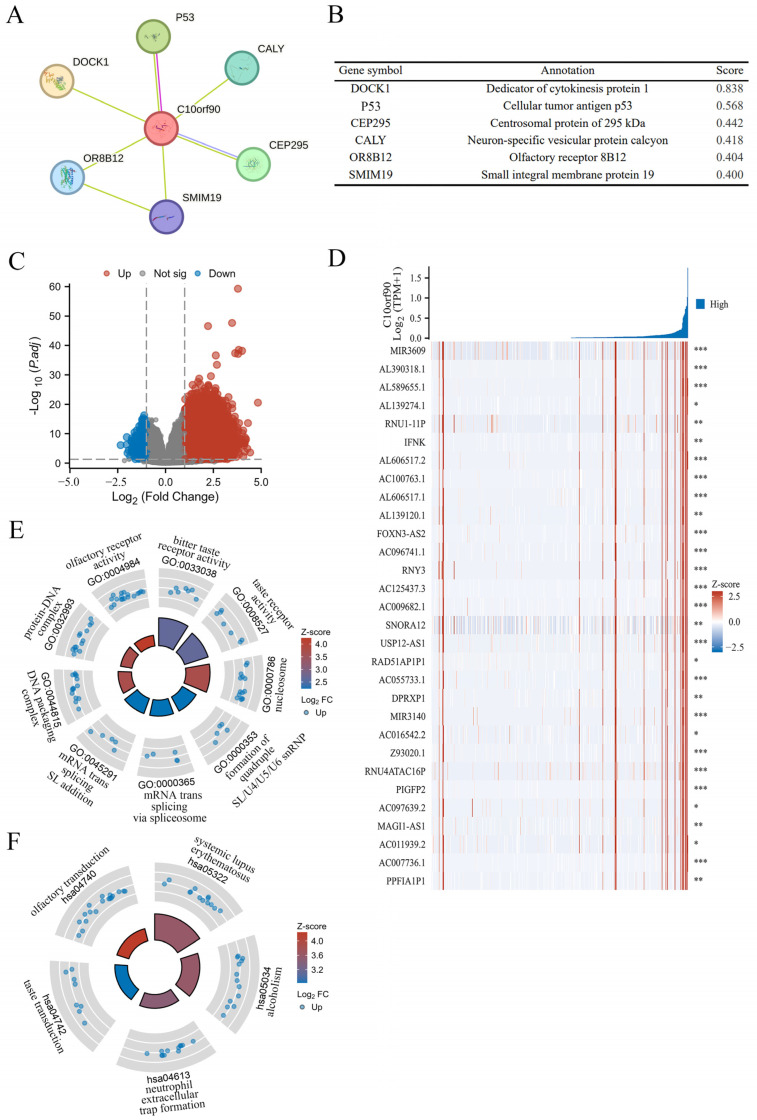
PPI network and gene function enrichment analysis of *C10orf90*. (**A**) The C10orf90 protein interacts with a PPI network. (**B**) The PPI network co-expression score. (**C**) A volcano plot of significantly correlated genes with *C10orf90* in COAD is presented. (**D**) Heatmap of the top 30 genes positively associated with the *C10orf90* gene in COAD. (**E**) GO enrichment analysis. (**F**) KEGG enrichment analysis. (* *p* < 0.05, ** *p* < 0.01, *** *p* < 0.001).

**Figure 10 ijms-25-10496-f010:**
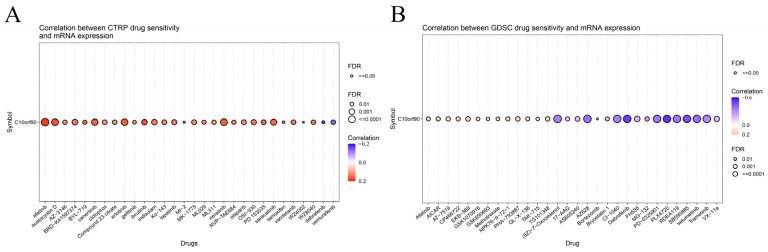
Sensitivity analysis of *C10orf90* related drugs. (**A**) Relationship between *C10orf90* and CTRP drug sensitivity. (**B**) Relationship between *C10orf90* and GDSC drug sensitivity.

**Figure 11 ijms-25-10496-f011:**
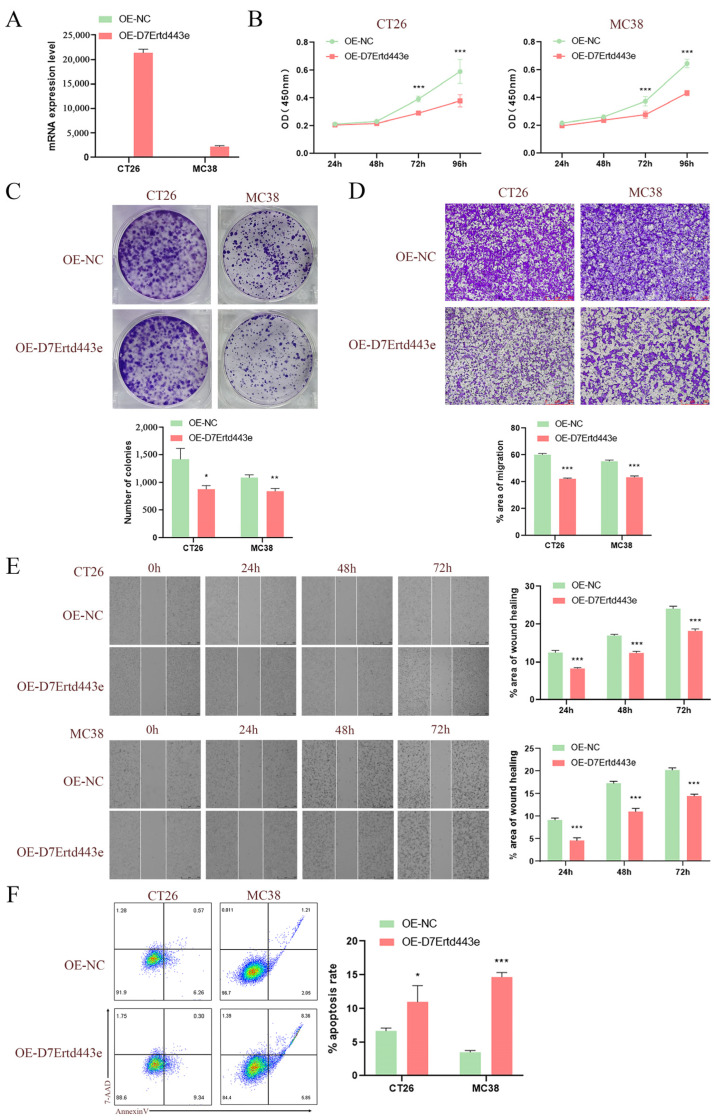
Overexpression of *C10orf90* inhibits colon cancer cell proliferation and tumor migration. (**A**) The mRNA expression level of *D7Ertd443e* in transfected cells. (**B**) CCK-8 assay of the impact of *D7Ertd443e* on tumor cell proliferation. (**C**) Plate cloning assay of the effect of *D7Ertd443e* on tumor cell proliferation. (**D**) Transwell assay of the impact of *D7Ertd443e* on tumor cell migration (scale: 250 μm). (**E**) Wound-healing assay of the effect of *D7Ertd443e* on tumor cell migration (scale: 500 μm). (**F**) Cell apoptosis assay of the effect of *D7Ertd443e* on tumor cell apoptosis. (Different colors represent cell density) (* *p* < 0.05, ** *p* < 0.01, *** *p* < 0.001).

## Data Availability

The data presented in this study are available in the main text, figures, tables, and Appendix A.

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
