# Peer review of "Novel Oncogenic Value of C10orf90 in Colon Cancer Identified as a Clinical Diagnostic and Prognostic Marker"

_ijms, 2024, doi:10.3390/ijms251910496_

Round 1

Reviewer 1 Report

Comments and Suggestions for Authors

Comments to authors

The work described in this manuscript shows a thorough bioinformatic investigation into the association of C10orf90 expression and cancer. However, there are multiple areas in the manuscript that need to be rewritten for clarity, especially the introduction. Also, there is not enough experimental detail to understand exactly what comparisons were being made and what the initial format of the data was used from the public databases.

Introduction

·       First paragraph of introduction is difficult to understand. Instead of providing two long sentences, please provide 3 or 4 clear, concise sentences. Additional editing of the entire Introduction is needed for clarity.

Results

·       Figure 1 needs to be redone or removed, not much value added

·       Figure 3 panel A, replace with GTEx by itself using same lollipop plot as B. Move panel C to supplemental or remove

·       Figure 4 panels A-C are not needed, should be Supplemental

·       Figure 9 panels E and F, should include actual names of pathways, not GO or hsa abbreviations

·       Table 1 needs to be moved to Supplemental

Methods

·       More detail is required in the Methods section, how was the fluorescence done in Figure 2, panel E

·       The statistical method used for many comparisons is not provided in detail

Discussion

·       In the survival analysis, increased C10orf90 expression is correlated with worse survival rates for many cancer types. However, for COAD, an increase in C10orf90 expression is correlated with reduced proliferation and migration and increased apoptosis. A short summary of why these opposing roles of C10orf90 may exist in disease outcomes should be included in the discussion.

Comments on the Quality of English Language

Sentence structure needs to be improved, especially in the Introduction

Reviewer 2 Report

Comments and Suggestions for Authors

Congratulations to the authors on the article they have presented.  It is a very comprehensive paper, but I suggest the following changes to improve their study:

- In Figure 1, change the word cancre to cancer.

- The colours that appear in Figures 2B, 2C and 2D, what are they supposed to indicate?  Also, the figures would look better if we could just see in which tissues C10orf90 expression mRNA appears depending on the base data. 

- Would it be possible to show the names of all the cancer types analysed? You have given the acronyms and it is difficult for a non-expert to understand.

- Figure 4A, B and C is difficult to see as it is very small. Is it possible to improve it?

- Cite the bibliographical references in the correct order.  In the Introduction the last reference is number 14, but in the Discussion section it is followed by reference 25 and in the Materials and methods section it starts with reference 15.  In addition, reference 33 is missing from the text.

- The conclusion could state in which tumour types the C10orf90 gene could be used as a diagnostic and prognostic biomarker. based on its results.
